# Development of *In Vivo* Haploid Inducer Lines for Screening Haploid Immature Embryos in Maize

**DOI:** 10.3390/plants9060739

**Published:** 2020-06-12

**Authors:** Chen Chen, Zijian Xiao, Junwen Zhang, Wei Li, Jinlong Li, Chenxu Liu, Shaojiang Chen

**Affiliations:** National Maize Improvement Center of China, College of Agronomy and Biotechnology, China Agricultural University, Yuanmingyuan West Road, Haidian District, Beijing 100193, China; b20173010014@cau.edu.cn (C.C.); xiaozj@cau.edu.cn (Z.X.); zhang1607168894@163.com (J.Z.); wellion@cau.edu.cn (W.L.); lijinlong306@163.com (J.L.)

**Keywords:** haploid induction rate, haploid immature embryo, haploid identification, maize

## Abstract

Doubled haploid technology is widely applied in maize. The haploid inducer lines play critical roles in doubled haploid breeding. We report the development of specialized haploid inducer lines that enhance the purple pigmentation of crossing immature embryos. During the development of haploid inducer lines, two breeding populations derived from the CAU3/S23 and CAU5/S23 were used. Molecular marker-assisted selection for both *qhir1* and *qhir8* was used from BC_1_F_1_ to BC_1_F_4_. Evaluation of the candidate individuals in each generation was carried out by pollinating to the tester of ZD958. Individuals with fast and clear pigmentation of the crossing immature embryos, high number of haploids per ear, and high haploid induction rate were considered as candidates. Finally, three new haploid inducer lines (CS1, CS2, and CS3) were developed. The first two (CS1 and CS2) were from the CAU3/S23, with a haploid induction rate of 8.29%–13.25% and 11.54%–15.54%, respectively. Meanwhile, the CS3 was from the CAU5/S23. Its haploid induction rate was 8.14%–12.28%. In comparison with the donor haploid inducer lines, the 24-h purple embryo rates of the newly developed haploid inducer lines were improved by 10%–20%, with a ~90% accuracy for the identification of haploid immature embryos. These new haploid inducer lines will further improve the efficiency of doubled haploid breeding of maize.

## 1. Introduction

Doubled haploid (DH) technology can reduce the time required for the development of inbred lines. Thus, it has been widely used in modern maize breeding programs [1]. One of the most important procedures in DH breeding is haploid induction, which is done by using special maize lines called the haploid inducer lines that can produce maternal haploids *in vivo* when pollinating to other germplasms. The first *in vivo* haploid inducer line, Stock6, was reported in 1959 and could produce 2%–3% haploids [2]. Modern haploid inducer lines such as CHOI1-3 [3], RWS [4], UH400 [5], PHI [6], and TAILS [7] have a haploid induction rate (HIR) of approximately 10% [8,9]. The HIR is a heritable trait controlled by multiple quantitative trait loci [5,10,11,12,13]. The genes located in two quantitative trait loci (QTLs), namely *qhir1* and *qhir8,* have been cloned and designated as *MTL/ZmPLA1/NLD* and *ZmDMP*, respectively [14,15,16,17]. The linkage genetic markers that flank these two genes are deemed as valuable information for marker-assisted selection (MAS) in haploid inducer breeding. The MAS of *qhir1* has proved to be effective in the development of haploid inducer lines CHOI3 [3] and 2GTAILs [7]. Since *ZmDMP* (*qhir8*) is a special gene that acts as the enhancer of *qhir1*, integrating these two genes in haploid inducer development could be more efficient [17], although only a few attempts have been made to use both *qhir1* and *qhir8* in the development of new haploid inducer lines. 

To achieve high-efficiency haploid production, both a high HIR and accurate haploid identification method are of equal importance for a new haploid inducer line. At present, haploid identification largely relies on the dominant gene, *R1-nj*. *R1-nj* leads to a colored aleurone layer owing to the xenia effect, but no pigmentation on the scutellum in the haploid [18]. Nevertheless, the influence of the female parent on the *R1-nj* expression resulted in a varied performance among germplasms [19]. Thus, it is critical to introduce other pigmentation-related markers for haploid identification. Except for the *R1* gene, *C1* [20], *Pl1* [21], and *P1* [22] have also been proved to control anthocyanin accumulation in kernel. Phenotypic selection for clearly discernable anthocyanin markers in both the scutellum and aleurone is important for producing haploid kernels. However, the weakly colored endosperm and shield of the seed capsule interfere in the identification of the haploid, thus selecting haploid embryo directly may be more efficient [1]. A fluorescence marker like green fluorescent protein (GFP) could be applied to identify haploid embryos [23]. In addition, an anthocyanin marker was also effective in selecting the haploid after an embryo culture *in vitro* [24]. Immature embryo was the optimal choice for its easy separation and regeneration [25]. Moreover, with the development of immature embryo identification techniques, the chromosome doubling of haploid immature embryos may further improve the DH efficiency.

Therefore, it is crucial to create a haploid inducer line with a stronger phenotypic marker to identify the haploid immature embryos from diploids. We addressed this need using a MAS for the *QTLs* of both *qhir1* and *qhir8* with subsequent selection based on the pigmentation of immature embryos. This approach yielded specialized haploid inducer lines with both a high HIR and an enhanced pigmentation of immature embryos.

## 2. Results

### 2.1. Validating the Effectiveness of MAS for qhir1 and qhir8

Nine genotype classes were obtained because of the combination of *qhir1* and *qhir8.* The plants from these genotype classes were crossed with ZD958, an elite hybrid used to evaluate the HIR. The results (Figure 1) revealed that the HIR of the plants that were absent at *qhir1* ranged from 0.70% to 1.04%, which was significantly lower than either those that carried a heterozygous *qhir1* allele or those that carried a homozygous *qhir1* from inducer, with HIRs of 3.77% to 5.27% and 10.02% to 14.42%, respectively. Moreover, the HIR increased significantly, 3%–14%, when the MAS was carried out with both *qhir1* and *qhir8*.

### 2.2. Selection for HIR and Number of Haploids per Ear (NHPE)

Two crosses, CAU3/S23 and CAU5/S23, that yielded 250 and 819 BC_1_F_1_ individuals, respectively, were used to develop the new haploid inducer lines. The MAS was used for *qhir1* and *qhir8*, and the individuals with a strong purple color in the immature embryos were chosen. Around 10% of the individuals were selected to generate the BC_1_F_2_ population (Figure 2B,D). Both the HIR and NHPE increased under the selection. The range of the HIR narrowed with each new generation (Figure 2A,C). In BC_1_F_5_, the HIR of the population derived from CAU5 ranged from 3.23% to 19.84%, and that derived from CAU3 ranged from 4.69% to 19.35% (Figure 2A,C). Even in the CAU3 population, the average HIR and NHPE values for the five candidate families reached 12.07% and 17.79, respectively. The HIR of BC_1_F_5_ was ~7.5% higher than that of BC_1_F_2_ (Figure 2B). The crosses yielded 11 candidate BC_1_F_5_ families from the CAU5 population, with average HIR and NHPE values of 10.28% and 14.96, respectively (Figure 2D). Ultimately, we obtained two candidate haploid inducer lines derived from CAU3/S23 namely CS1 and CS2, with an average HIR of 10.62% and 11.54%, respectively, and one candidate haploid inducer line derived from CAU5/S23, namely CS3 with an average HIR of 11.71%. 

### 2.3. Relationship between the NHPE, HIR, and Number of Kernels per Ear (NKPE) 

The individuals from BC_1_F_2_ to BC_1_F_5_ were evaluated with three indexes, NHPE, HIR, and NKPE. Figure 3 presents the relationship between each pairing of the three parameters. The NHPE showed a significant positive correlation with the HIR in both populations (Figure 3A,D). After we chose the top 10% of the NHPE as a threshold, the individuals from the CAU3 population produced more than 23 haploids, and those from the CAU5 population produced more than 26 haploids per ear. High NHPE yield was found to be in the NKPE range from 100 to 300 (Figure 3B,D). Nevertheless, after we used the top 10% of the HIR, the individuals from the CAU3 population showed a HIR higher than 14.88%, and those from the CAU5 population showed a HIR above 15.11%, and the optimal NKPE was 0–300. Therefore, to achieve a high yield of the haploids, the best selection of NKPE may be 100–300.

### 2.4. Evaluating the HIR among the Different Varieties

Evaluation of the candidate lines with ZD958 suggested that no significant difference was observed in the HIR between the newly developed haploid inducer lines and the corresponding donor inducer (Table 1). However, significant differences between the newly developed haploid inducer lines and donor parents were observed for NHPE, NKPE, and EmAR. As listed in the table below, the NHPEs of CS1 and CS2 were 20.71 ± 1.45 and 33.93 ± 3.42, respectively. These values were significantly higher than those of CAU3. A similar result was found for the NKPE of CS1 and CS2. Comparing the NHPE and NKPE of CAU5 (17.00 ± 2.70 and 131.64 ± 12.82, respectively) and CS3 (33.18 ± 3.39 and 282.45 ± 17.56, respectively), the newly developed haploid inducer line CS3 had a significantly improved NHPE and NKPE. The EmAR was significantly improved during the breeding of CS1 and CS3, but significantly decreased in the breeding of CS3. The evaluation of the newly developed haploid inducer lines with five more commercial hybrids revealed that for CS1, the HIR range was 8.29%–13.25%, and the NHPE range was low, 13.00–24.71, owing to the lower NKPE range of 138.22–196.53. For CS2, the NHPE and HIR were in the range of 26.63–39.18 and 11.70%–15.54%, respectively. CS3 had 18.21–34.42 haploids per ear and an HIR range of 8.14%–12.28%. These results showed that three candidate haploid inducer lines could achieve high haploid yields in a different genetic background. 

### 2.5. Evaluation of the Pigmentation in Crossing Embryos and Identification of the Haploid Embryos

The lines where the crossing embryos had a color score of 4 to 5 were selected as the candidate haploid inducer lines to evaluate the deepening of the pigmentation (Figure 4A). For the BC_1_F_3_ individuals, the average 24-h purple embryo rate of the CAU3 and CAU5 populations was 84.5% and 65.9%, respectively. Significant improvement was noted in the CAU3 and CAU5 populations of the BC_1_F_4_ generation (by 11.6% and 20.4%, respectively) (Figure 4B). In the ZD958 ears pollinated by the three candidate haploid inducer lines, the putative candidate haploid embryos that lacked pigmentation were selected randomly and verified by polymorphic molecular markers. Among the 68 embryos from ZD958/CAU3, 7 were diploid. Among the 96 putative haploids from ZD958/CS1, 8 were diploid. Among the 134 putative haploids of ZD958/CS2, 6 were diploid. The accuracy of the haploid identification using CAU3 was 89.71%, which was lower than that of CS1 (91.67%) and CS2 (95.52%). Similarly, the accuracy for CS3 improved by 1.7% compared to that of CAU5.

### 2.6. Agronomic Traits

In comparison to the corresponding donor parents, three candidate haploid inducer lines demonstrated superior agronomic performance in terms of plant height, ear height, number of tassel branches, tassel size, and NKPE (Table 2). The number of tassel branches for CS1 and CS2 was 3-fold and 4-fold higher than that of CAU3, respectively. CS3 had a ~3-fold greater number of tassel branches than CAU5. The tassel size score (see Figure 4) decreased as the number of tassel branches increased. The fecundity of the newly developed haploid inducer lines was much greater than that of the donor haploid inducer lines, as the CS1 and CS2 self-pollination ears produced 37 and 54 more seeds on average, respectively, than CAU3. The average NKPE for CS3 was 110.6, which was much greater than that for CAU5 (Table 2). Finally, with regard to the days to anthesis, no obvious differences were observed between the candidate haploid inducer lines and the donor lines.

## 3. Discussion

### 3.1. MAS for Multiple Loci Related to Haploid Induction

The effectiveness of MAS for *qhir1* has been reported, although the HIR is only ~2% [3,7]. A simultaneous selection of the two loci was used in our study. This was a more efficient method for producing haploid inducer lines. We used the CAU6 NILs to exclude the influence of genetic background and accurately evaluate the contributions of MAS based on *qhir1* and *qhir8*. We revealed that the combination of the homozygous *qhir1* and *qhir8* yielded a superior HIR compared to other genotype classes. In haploid inducer line breeding programs, application of the MAS for *qhir1* and *qhir8* could potentially eliminate ~90% of low-HIR individuals, which would significantly reduce the time and efforts needed for HIR testing. However, we found that there was still substantial variation (0%–8%) with respect to the HIR among the individuals that were homozygous on both *qhir1* and *qhir8*, suggesting that other genes in addition to *ZmPLA1* and *ZmDMP* also play very important roles in regulating the HIR. MAS for additional quantitative trait loci or genes that contribute to haploid induction may provide more efficient methods. The use of MAS to maintain the haploid induction allele could improve selection efficiency as well as save time and resources that are required for haploid inducer line development. 

### 3.2. Development of an Efficient Haploid Inducer Lines

The efficiency of DH breeding has increased significantly with improvements in haploid induction, haploid identification, and chromosome doubling. In terms of haploid induction, many haploid inducer lines have been developed with a high HIR, and additional marker systems that are distinct from *R1-nj* can be used for different environments, germplasms, and stages of haploid identification. An oil-content marker also has been integrated into many maternal haploid inducer lines, and automatic identification of haploids has been accomplished based on the high oil inducer lines [3,26,27]. In addition, indicators such as red roots [6] and a purple sheath [28,29] have been integrated into haploid inducer lines to improve the efficiency of haploid identification at the seedling stage. The application of these markers can provide additional confirmation for candidate haploids and eliminate the false-positive ones. All haploid inducer lines carry *R1-nj*, which has been the most widely used marker for identifying haploids accurately and efficiently. The sorting of haploid immature embryos can differ significantly among haploid inducer lines of different backgrounds. *R1-nj* expression is controlled by genes encoding structural proteins involved in anthocyanin biosynthesis, such as *A1*, *A2*, *C2*, *Bz1*, and *Bz2,* as well as the regulatory genes *C1/Pl* and *WD40* [30,31]. Other genes that regulate anthocyanin production to increase *R1-nj* expression and reduce the time needed for maximal expression may be employed to identify haploid immature embryos. In this study, the pigmentation and HIR were assessed in each generation until the candidate haploid inducer lines were obtained. The efficient identification of the haploid immature embryos via an enhanced performance of *R1-nj* expanded the application of the anthocyanin markers. *R1-nj* is more efficient and intuitive compared to methods based on a fluorescent protein [23,32]. Thus, the use of *R1-nj* marker offers the possibility of easily producing an abundance of haploid immature embryos and establishing technologies for identifying haploid immature embryos and assessing chromosome doubling.

## 4. Materials and Methods 

### 4.1. Materials 

Two *in vivo* haploid inducer lines, CAU3 and CAU5, and one elite inbred line, S23, were used to develop the two breeding populations. Both CAU3 and CAU5 were developed by China Agricultural University, with different characteristics: CAU3 has deep-purple leaves and stems, and CAU5 has green leaves and stems. Both CAU3 and CAU5 had an HIR of ~10%. The S23 was inbred with lines that have *R1-nj*, the marker for the purple plumule, and a big tassel. 

Six commercial hybrids were used as testers to evaluate both the HIR and the efficiency of identifying the haploid immature embryos. The hybrid Zhengdan958 (ZD958) was applied in each generation, and the other five hybrids (DK653, Jingke968 (JK968), Nongda372 (ND372), Nongda678 (ND678), and Xianyu335 (XY335)) were used as testers for the evaluation of the candidate haploid inducer lines. Field planting was conducted at two locations, Beijing and Hainan, between 2015 and 2019.

### 4.2. Breeding Scheme

F_1_ seeds were obtained from crosses of CAU3 with S23 and CAU5 with S23. F_1_ hybrids were backcrossed with the corresponding haploid inducer lines to generate the BC_1_F_1_ populations, and then self-pollinated until BC_1_F_5_ was generated. MAS for both *qhir1* and *qhir8* was carried out from BC_1_F_1_ to BC_1_F_4_ with the linkage markers X291 and Chr9-76 [11,12]. Meanwhile, individuals from BC_1_F_2_ to BC_1_F_5_ were evaluated for five objective traits, namely, HIR, NHPE, NKPE, immature embryo color score, and agronomic performance, to determine the most suitable candidate in each generation. The HIR was calculated as follows: HIR = NHPE/NKPE × 100%.

### 4.3. Verification of the MAS for qhir1 and qhir8 during the Selection for HIR 

The MAS for *qhir1* and *qhir8* was assessed in self-crossed or hybrid progenies of four near-isogenic lines (NILs), namely, CAU6^(*ZmPLA1-ZmDMP*)^, CAU6^(*ZmPLA1-zmdmp*)^, CAU6^(*zmpla1-ZmDMP*)^, and CAU6^(*zmpla1-zmdmp*)^, in a B73 background [17]. The lowercase gene name *zmpla1/zmdmp* in superscript means the existence of an allele from the haploid inducer line. Conversely, those in capital form indicate alleles from B73. Nine genotype groups were classified according to the combination of *qhir1* and *qhir8* by the PCR of the polymorphic molecular markers *qhir1* (5‘–3’, F: GTGCCACCCACTCTTCTTCA, R: ACTGTCACTTCCCACCGTCA) and *qhir8* (5‘–3’, F: CACACGTCAGTGCAGGAAAT, R: AGTCGTTGCTGCCTCTCAGT) [33,34]. To evaluate the HIR, more than 20 crosses with ZD958 were performed for each genotype.

### 4.4. Haploid Identification Based on the Pigmentation of Crossing Embryos

Immature embryos from crossing ears were stripped at 15 days after pollination, as described by Fontanet and Vicient [25]. The embryos were then immediately cultured in Murashige and Skoog medium (3 g/L, with sucrose 30 g/L; agar 7.5 g/L; and pronamide 2 µM, pH 5.8). After 24 h, the embryo color was assessed on a five-point scale (see below). Hybrid progeny embryos were purple, whereas haploid embryos were colorless.

The embryo color scores were assigned according to the total area that was colored and the intensity of the color of the scutellum. A score of 1 indicated a small area of color and low intensity in the crossing embryos, whereas embryos with strong color were given a score of 5. Haploid inducer lines with a color score of 4 or 5 were assessed again in each subsequent generation. For BC_1_F_3_ and BC_1_F_4_, the number of purple embryos for each set of crossing embryos after 24 h of culture was also recorded. The 24-h purple embryo rate was calculated as follows: 24-h purple embryo rate = (the number of purple embryos at 24 h/total number of purple embryos) × 100%. To evaluate the accuracy of haploid identification by candidate haploid inducer lines, candidate haploid embryos were selected randomly and verified by the polymorphic molecular markers between the haploid inducer lines and testers. True haploids did not show the electrophoretic band that was present in the haploid inducer lines.

### 4.5. Assessment of the Agronomic Performance

The three candidate high-anthocyanin haploid inducer lines CS1, CS2, and CS3 and the two parent haploid inducer lines CAU3 and CAU5 were evaluated for six agronomic traits: days to anthesis, plant height, ear height, number of tassel branches, NKPE, and tassel size (1–5). Tassel size were scored as described by Chaikam et al. [7]; a score of 1 represents large tassels, whereas a score of 5 represents small tassels.

### 4.6. Statistical Analysis

All statistical analyses were carried out using GraphPad Prism 8.5 and Excel 2016. 

## 5. Conclusions

Three haploid inducer lines were developed by MAS for both *qhir1* and *qhir8*, and phenotypic selection based on HIR, NHPE, and purple embryo rate. We highlighted that the MAS on both *qhir1* and *qhir8* and phenotypic selection on NHPE were beneficial for achieving a high haploid yield. The candidate haploid inducers obtained in this study, CS1, CS2, and CS3, showed an enhanced 24-h purple embryo rate and were higher in NHPE compared to the donor parents. These new haploid inducer lines could further improve the efficiency of haploid induction and the identification of immature embryos.

## Figures and Tables

**Figure 1 plants-09-00739-f001:**
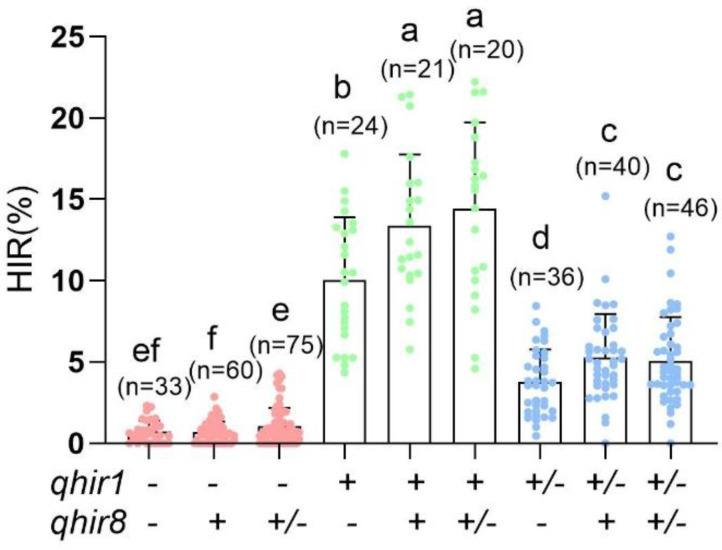
Haploid induction rate (HIR) values for the CAU6 NILs (near-isogenic lines) for nine genotypes. Different lowercase letters indicate a significant difference at *p* < 0.05 (Tukey’s honestly significantly different (HSD) test). (+): presence of homozygous *qhir1/qhir8* allele from inducer lines; (+/-): presence of heterozygous *qhir1/qhir8* allele; (-): presence of homozygous allele at *qhir1/qhir8* from non-inducer line; n: number of ears used for calculating the HIR of each genotype.

**Figure 2 plants-09-00739-f002:**
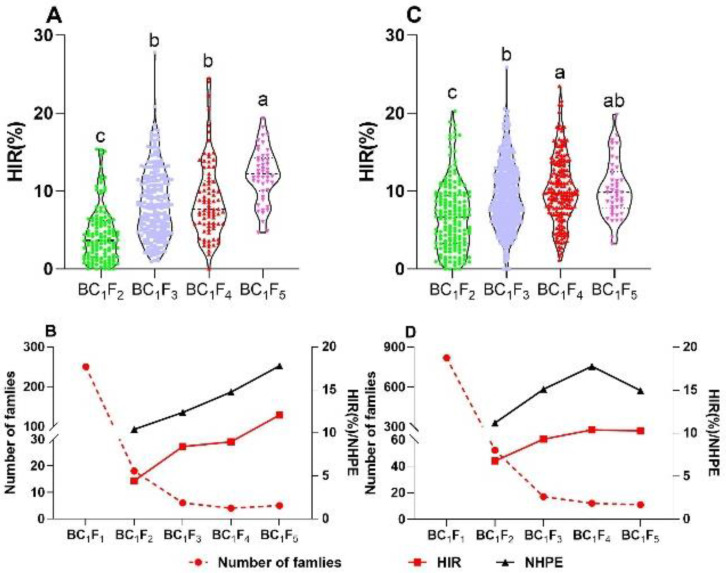
Selection response on the number of families, HIR, and number of haploids per ear (NHPE) for the different generations. More than three numbers of ears were used to calculate the HIR and NHPE, and a Tukey’s honestly significantly different (HSD) test was performed to identify the difference among the different selection generations. (**A**) and (**B**) correspond to the population of CAU3; (**C**) and (**D**) correspond to the population of CAU5.

**Figure 3 plants-09-00739-f003:**
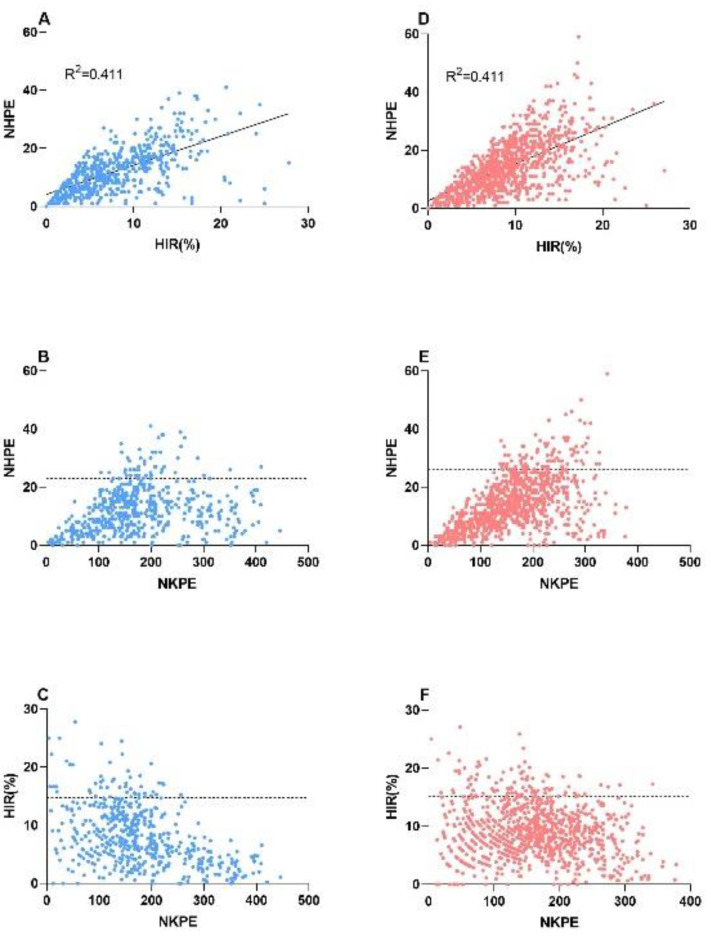
Analysis of the relationships between the NHPE, HIR, and NKPE. (**A**)–(**C**) correspond to the population of CAU3; (**D**)–(**F**) correspond to the population of CAU5. The solid lines in (**A**),(**D**) indicate the trend of the correlation analysis. The dotted lines on (**B**),(**C**),(**E**),(**F**) represent the threshold of the top 10% of the NHPE/HIR.

**Figure 4 plants-09-00739-f004:**
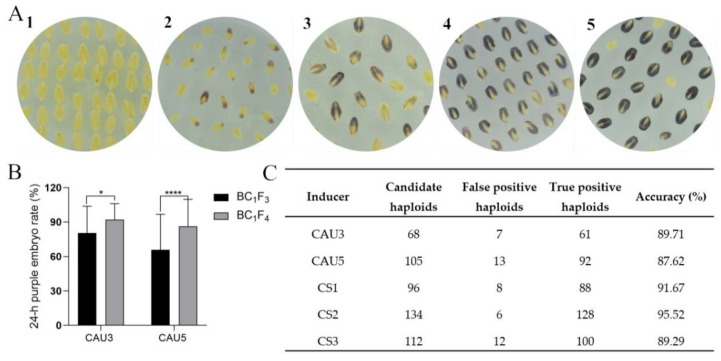
(**A**) Classifying the color of immature embryos. (**B**) Comparison of the 24-h purple embryo rate of crossing embryos between BC_1_F_3_ and BC_1_F_4_. * *p* < 0.05, **** *p* < 0.0001. (**C**) Efficiency of identifying haploid immature embryos.

**Table 1 plants-09-00739-t001:** Evaluation of the candidate haploid inducer lines.

Inducer	Tester	Total Number	NHPE	NKPE	EmAR (%)	HIR (%)
CAU3	ZD958	475	10.00 ± 3.11	95.00 ± 26.89	3.14 ± 1.42	11.22 ± 1.79
CAU5	ZD958	1843	17.00 ± 2.70	131.64 ± 12.82	10.96 ± 0.50	11.89 ± 0.77
CS1	ZD958	3341	20.71 ± 1.45 **	196.53 ± 7.56 ****	6.72 ± 0.67 *	10.62 ± 0.68
CS2	ZD958	4059	33.93 ± 3.42 **	289.93 ± 16.44 ****	6.31 ± 0.59 *	11.54 ± 0.87
CS3	ZD958	3107	33.18 ± 3.39 **	282.45 ± 17.56 ****	5.47 ± 0.91 **	11.71 ± 1.01
CS1	DK653	1358	24.57 ± 2.64	194.00 ± 19.95	5.35 ± 0.76	12.62 ± 0.61
	JK968	993	19.33 ± 3.73	165.50 ± 22.29	7.67 ± 1.29	11.56 ± 1.16
	ND372	1244	16.89 ± 2.34	138.22 ± 10.20	9.34 ± 1.87	11.97 ± 1.07
	ND678	1324	24.71 ± 3.04	189.14 ± 11.84	8.77 ± 1.11	13.25 ± 1.57
	XY335	439	13.00 ± 4.93	146.33 ± 39.08	5.56 ± 0.84	8.29 ± 1.35
CS2	DK653	3167	29.93 ± 3.49	211.13 ± 17.20	5.68 ± 0.72	13.72 ± 0.78
	JK968	1612	26.63 ± 5.37	201.50 ± 27.60	7.54 ± 1.04	12.59 ± 1.46
	ND372	2648	27.42 ± 2.01	220.67 ± 10.67	9.61 ± 0.61	12.49 ± 0.84
	ND678	2741	30.00 ± 3.18	195.79 ± 16.20	6.86 ± 0.69	15.54 ± 1.00
	XY335	3685	39.18 ± 3.25	335.00 ± 20.03	5.46 ± 1.01	11.70 ± 0.68
CS3	DK653	2933	25.86 ± 2.78	209.50 ± 14.41	4.35 ± 0.59	12.27 ± 0.94
	JK968	3674	31.90 ± 2.37	367.40 ± 13.62	2.88 ± 0.47	8.66 ± 0.54
	ND372	3762	23.77 ± 2.37	289.38 ± 16.35	5.11 ± 0.64	8.14 ± 0.58
	ND678	2135	18.21 ± 2.26	152.50 ± 16.15	4.24 ± 0.91	11.82 ± 0.73
	XY335	3340	34.42± 3.62	278.33 ± 16.31	5.87 ± 1.03	12.28 ± 0.99

* *p* < 0.05, ** *p* < 0.01, *** *p* < 0.001, **** *p* < 0.0001, for comparing the values between the parent haploid inducer line and the candidate haploid inducer lines (two-tailed *t* test).

**Table 2 plants-09-00739-t002:** Performance of the agronomic traits among the three candidate haploid inducer lines compared with CAU3 and CUA5.

Inducer	Days to Anthesis	Plant Height (cm)	Ear Height (cm)	Tassel Branches Number	Tassel Size Score	Number of Kernels per Ear
CAU3	66.40 ± 0.60	174.80 ± 1.43a	50.60 ± 1.55	5.80 ± 0.25	4.00 ± 0.00	49.00 ± 2.34
CAU5	59.80 ± 0.72	149.00 ± 1.13	32.60 ± 0.82	9.80 ± 0.86	3.20 ± 0.13	37.20 ± 3.16
CS1	67.80 ± 0.37	196.80 ± 6.26 *	94.40 ± 3.08 ****	9.20 ± 0.80 **	3.20 ± 0.20 **	86.00 ± 8.64 **
CS2	66.60 ± 0.75	198.00 ± 3.15 ***	82.60 ± 1.21 ****	9.20 ± 0.37 ***	3.00 ± 0.00 **	103.00 ± 7.46 ***
CS3	62.00 ± 0.32	162.80 ± 2.24 ***	45.80 ± 2.01 ***	28.20 ± 1.11 ****	1.20 ± 0.20 ***	110.60 ± 3.44 ****

* *p* < 0.05, ** *p* < 0.01, *** *p* < 0.001, **** *p* < 0.0001, for comparing values between the parent haploid inducer line and the candidate haploid inducer lines (two-tailed *t* test).

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
