# Peer review of "Development of In Vivo Haploid Inducer Lines for Screening Haploid Immature Embryos in Maize"

_plants, 2020, doi:10.3390/plants9060739_

Round 1

Reviewer 1 Report

This is a review of the manuscript submitted by Chen et al. to Plants. In this paper, the authors present how they developed haploid inducer lines for screening immature haploid embryos in maize. The manuscript was prepared in a very solid way and would bring a big impact on researchers interested in doubled haploid technology. In my opinion, the review can be published in the current form with minor improvements.

I have the following suggestions:

My suggestions and questions concern mostly statistics.

Line 239: Please specify what type of statistical analysis did you perform to compare the data. In some places, it looks like ANOVA and some post hoc test (Tuckey’s or Duncan’s test).

Figure 1. Please specify under the description of the Figure how many individuals did you evaluate. The threshold for statistical significance should be p ≤ 0.05. Please write what post-hoc test dis you use. The letters (“a”, “b”, “c”, etc) should be added in the following order: “a” – for the highest mean value in the data set. By the letter "a" samples 5 and 6 should be marked.

Figure 2. Please specify under the description of the Figure how many individuals did you evaluate. Would it be possible to add the statistics?

Figure 3. A and D, contain trend-line. Please add this to the description. How many individuals did you evaluate?

Figure 4. A. I suggest changing a-e to 1-5 (color score classes). C. Would it be possible to add the statistics?

Table 1. For NHPE and NKPE, EmAR (%), and HIR(%), please try to change the statistics ANOVA + post hoc test.

Table 2. The same remark as for Table 1.

Author Response

Comment 1: page 2, lines 61. Number of Individuals, threshold statistical significance, type of post-hoc test and markers in Figure1.

Response: Thank you for your suggestion. The number of ears used for calculating the HIR of each genotype has been added. Tukey’s Honestly Significantly Different (HSD) test was used to test the significant difference at p < 0.05, it has been added in line 61.

Comment 2: page3, lines 81. The number of individuals for evaluation and the possible of add statistics.

Response: Thank you for your advice. More than three tester ears were used to calculate HIR and NHPE for each individual, a Tukey’s Honestly Significantly Different (HSD) test was performed to identify the difference among different selection generations. Please refer to line 81-82.

Comment 3: page3, lines 97. Description of trend line and the number of individuals for evaluating.

Response: Thank you for your comments. The solid line in Figure 3A and Figure 3D indicates trend of correlation analysis. The dotted line in Figure 3B, Figure 3C, Figure 3E and Figure 3F represents the threshold of the top 10% of NHPE/HIR. Each spot denotes one ears of HIR evaluation, the individuals from BC1F2 to BC1F5 were used in Figure 3 and the number has been shown in Figure 2, Please refer to the revisions in the manuscript line N to N and Legend of Figure N.

Comment 4: page5, lines 133. A. changing a-e to 1-5 (color score classes). C. possible to add the statistics?

Response: Thank you for your comments. As the pigmentation of R1-nj in immature changes from low to the high density, the scale categorized by number is more reasonable. So we changed a-e to 1-5 as suggested. However, in this study, the color score was classified by visual score, so it is not appropriate to do statistical analysis. nevertheless, your valuable suggestion remind us that  this analysis would be done by image processing techniques in the future.  C: Accuracy of haploid identification was only a single data to do a simple comparison, which have no need for statistics.

Comment 5: page4, lines 118 and page5, lines 148.Try to change the statistics ANOVA + post hoc test for Table 1 and Table 2.

Response: Thank you for your valuable comment and advice. In Table 1 and Table 2, values between parent haploid inducer line and candidate haploid inducer lines were compared, include CAU3 vs CS1, CAU3 vs CS2 and CAU5 vs CS3. Significance test was carried out through two-tailed t test.

Reviewer 2 Report

Dear Editor,

In the presented study, authors showed efficient method of the development of haploid inducer line in maize with phenotypic marker enabling accurate haploid identification. However, the manuscript should be prepared more carefully. Some sentences/part of manuscript are not clear. Especially abstract should be written more clearly. At present it is difficult to understand the objective, method, and result. Figure2 is not sufficiency explained. There are some editorial errors eg. In Abstract (line17) and Materials and Methods in breeding scheme (lines 200) it is said that BC1F1 populations was self-pollinated until BC1F4, whereas in Results, Section 2.2 the authors wrote about BC1F5 population. BC1F5 population was also shown on Figure 2. The manuscript could be suitable for publication in Plants after revision.

Author Response

Response: Thank you for your valuable comment and advice. We have checked and corrected mistakes in the manuscript. The manuscript has been carefully read by all authors and revised the sentences that we think not clear. Please refer to revisions of abstract and line 12-24, 32-52, 56, 63-74, 88-96, 99-113, 121-127,138-146, 167, 178-179, 194, 201-203 and 208-209.